# Categorial Grammar Induction as a Compositionality Measure for Emergent Languages in Signaling Games

**Ryo Ueda[1], Taiga Ishii[1], Koki Washio[2]\*, & Yusuke Miyao[1]**
[1]The University of Tokyo
[2]Megagon Labs, Tokyo
[1]{ryoryoueda,taigarana,yusuke}@is.s.u-tokyo.ac.jp
[2]kwashio@megagon.ai

## Abstract

This paper proposes a method to investigate the syntactic structure of emergent languages using categorial grammar induction. Although the structural property of emergent languages is an important topic, little has been done on syntax and its relation to semantics. Inspired by previous work on CCG induction for natural languages, we propose to induce categorial grammars from sentence-meaning pairs of emergent languages. Since an emergent language born in a signaling game is represented as pairs of a message and meaning, it is straightforward to extract sentence-meaning pairs to feed to categorial grammar induction. We also propose two compositionality measures that are based on induced grammars. Our experimental results reveal that our measures can recognize compositionality. While correlating with existing measure TopSim, our measures can gain more insights on the compositional structure of emergent languages from induced grammars.

## 1 Introduction

Despite its importance, few methods have been established to evaluate the structure of emergent languages with respect to *syntax* and *semantics*. Previous work frequently employs a *signaling game* (Lewis, 1969) or its variant, where the agents are a mapping from *a meaning space* to *a message space* or its inverse. The problem is that little has been analyzed on how syntax combines messages to yield semantics or meanings. Such a structural property is known as *compositionality*.

To analyze the syntax of emergent languages, we focus on categorial grammar induction (CGI, e.g., Zettlemoyer & Collins, 2005) and propose to apply it to emergent languages. Since CGI derives a lexicon and a semantic parser given sentence-meaning pairs, it is suitable for the syntactic analysis of a language emerging as message-meaning pairs in a signaling game. We also propose compositionality measures built on the F1-score for unseen data and the lexicon size of CGI parsers. It is based on intuition that a compositional language is expected to be generalized and described by a minimal lexicon.

Compositionality measures have been proposed for emerging languages, such as topographic similarity (TopSim, Brighton & Kirby, 2006), tree reconstruction error (TRE, Andreas, 2019), positional disentanglement (PosDis, Chaabouni et al., 2020), and bag-of-symbols disentanglement (BosDis, Chaabouni et al., 2020). We choose TopSim to compare with ours, since it is the most popular in this area (e.g., Lazaridou et al., 2018).

Pioneering and suggestive work by van der Wal et al. (2020) on the syntax of emergent languages proposes to apply unsupervised grammar induction (UGI) originally developed for natural languages: CCL (Seginer, 2007) and DIORA (Drozdov et al., 2019). UGI is reasonable if neither gold derivations nor meanings are available. Note that UGI estimates the structure of emergent languages given only messages, whereas ours is intended to derive not only the structure but also the systematic composition of messages to meanings given message-meaning pairs.

---

\*Work done at the University of Tokyo.

Our contributions are (1) to propose to apply categorial grammar induction (CGI) to emergent languages for understanding their structure, (2) to propose two CGI-based compositionality measures that are more syntax-aware than existing compositionality measures, and (3) to show they can indeed measure compositionality.

## 2 CATEGORIAL GRAMMAR INDUCTION

In this section, we introduce categorial grammar (CG), CG-based semantic parsing, and its induction (CGI) for natural languages[1]. CGI is also eligible for emergent languages in signaling games, as it derives a lexicon and a parser from message-meaning pairs. Note that semantic parsing means a conversion of messages into the corresponding meanings.

Figure 1: Example derivation tree of "look left 1" by categorial grammar.

### 2.1 CATEGORIAL GRAMMAR

The formalism for our semantic parsing is *categorial grammar* (CG, Steedman, 1996; 2000). A lexical entry $w \vdash X : \psi$ is a triple of a word $w$, a category $X$ (defined below), and a logical form $\psi$. Consider the following example pair of a message and its logical form: $\langle$"look left 1", $\texttt{iter}(\texttt{and}(\texttt{lturn}, \texttt{look}), 1)\rangle$. Their lexical entries can be described as follows:

$$\texttt{look} \vdash \texttt{V} : \texttt{look}, \quad \texttt{left} \vdash \texttt{S}\backslash\texttt{V} : \lambda x.\texttt{and}(\texttt{lturn}, x), \quad \texttt{1} \vdash \texttt{S}\backslash\texttt{S} : \lambda x.\texttt{iter}(x, 1).$$

Symbols such as $\texttt{V}$ and $\texttt{S}\backslash\texttt{V}$ represent *categories*. A category is either an atomic category of the form $\texttt{N}$, $\texttt{V}$, or $\texttt{S}$, or a complex category of the form $X/Y$ or $X\backslash Y$ where $X, Y$ are categories. The atomic categories $\texttt{N}$, $\texttt{V}$, and $\texttt{S}$ stand for the linguistic notions of noun, intransitive verb, and sentence respectively[2]. In addition, CGs have *application rules* to describe the way to combine adjacent categories.

$$X/Y : f \quad Y : a \quad \Rightarrow \quad X : f(a) \tag{>}$$
$$Y : a \quad X\backslash Y : f \quad \Rightarrow \quad X : f(a) \tag{<}$$

where $X, Y$ are categories. The first rule named ">" is called the *forward application rule*, while the second rule named "<" is called the *backward application rule*. Rule $>$ (resp. $<$) means that a predicate $f$ of category $X/Y$ (resp. $X\backslash Y$) can take an argument $a$ of category $Y$ to yield $f(a)$ of category $X$. With the lexical entries and the application rules, we can construct a derivation tree of "look left 1" as shown in Figure 1.

### 2.2 LOG-LINEAR PROBABILISTIC CGS AND CG INDUCTION

Given a set of lexical entries $\Lambda$, there might be multiple derivations for each message. Following previous work (e.g., Zettlemoyer & Collins, 2005), we choose the most likely derivation by using a *log-linear model* that contains a feature vector function $\phi$ and a parameter vector $\theta$. Given a message $m$, the joint probability of a logical form $\psi$ and a derivation $\tau$ is defined as:

$$P(\tau, \psi \mid m; \theta, \Lambda) = \frac{e^{\theta \cdot \phi(m, \tau, \psi)}}{\sum_{(\tau', \psi')} e^{\theta \cdot \phi(m, \tau', \psi')}}.$$

Then, *semantic parsing* is a problem to find the most likely logical form $\hat{\psi}$ given $m$:

$$\hat{\psi} = \arg \max_\psi P(\psi \mid m; \theta, \Lambda) = \arg \max_\psi \sum_\tau P(\tau, \psi \mid m; \theta, \Lambda).$$

Thus far, several studies have proposed methods for *CG induction*, the task of which is to find a suitable $\Lambda$ and $\theta$ from a given set of message-meaning pairs $\{(m, \psi)\}$ (e.g., Zettlemoyer & Collins, 2005; Kwiatkowski et al., 2010; Artzi et al., 2014). The induction algorithm updates $\Lambda$ and $\theta$ so that $\sum_{(m, \psi)} \log P(\psi \mid m; \theta, \Lambda)$ is maximized.

---

[1] Although previous work is on combinatory categorial grammar (CCG), we restrict it to CG.

[2] The category of intransitive verbs is usually $\texttt{S}/\texttt{N}$ ($\texttt{S}/\texttt{NP}$) or $\texttt{S}\backslash\texttt{N}$ ($\texttt{S}\backslash\texttt{NP}$), but we regard $\texttt{V}$ as an atomic category. It is because the languages we define in Section 4.1 take an imperative form without any subject.

## 3 CGI AS A COMPOSITIONALITY MEASURE

We propose two compositionality measures CGF and CGL, which are based on an induced categorial grammar. Let $\mathcal{E}_{\text{train}}, \mathcal{E}_{\text{test}}$ be a training and test data for CGI. We train a log-linear model with $\mathcal{E}_{\text{train}}$ to derive a lexicon $\Lambda$ and a parameter $\theta$ and test it with $\mathcal{E}_{\text{test}}$ to calculate the F1-score for semantic parsing. Here, precision is defined as #correctly parsed/#parsed, while recall is defined as #correctly parsed/$|\mathcal{E}_{\text{test}}|$ (Zettlemoyer & Collins, 2005). Then, CGF and CGL are defined as:

$$\text{CGF} = \text{F1-score}, \quad \text{CGL} = |\Lambda|$$

The higher CGF (resp. lower CGL) is, the more compositional a language is judged, since a compositional language is expected to be generalized for the communication of unseen data and described by a minimal lexicon.

## 4 EXPERIMENTAL SETUP

### 4.1 SIGNALING GAME

**Input Space** We define two input spaces for our signaling game: *Lang-attval* and *Lang-conj* [3]. Lang-attval is the same as attribute-value inputs (e.g., Kottur et al., 2017), while Lang-conj is more complex. First, *Lang-attval* is defined as the set of sequences derived from the following context-free grammar with a start symbol S:

$$\text{S} \rightarrow \text{V}' \text{ R} \quad \text{V} \rightarrow \text{look} \mid \text{jump} \mid \text{walk} \mid \text{run}$$
$$\text{V}' \rightarrow \text{V D} \quad \text{D} \rightarrow \text{left} \mid \text{right} \mid \text{up} \mid \text{down} \quad \text{R}' \rightarrow 1 \mid 2 \mid 3 \mid 4$$

Next, let S″ be a start symbol. Then, *Lang-conj* is the set of sequences derived from the above context-free grammar in addition to the following rules:

$$\text{S}'' \rightarrow \text{ S} \mid \text{S S}' \quad \text{S}' \rightarrow \text{ and S}$$

**Game Procedure** In our signaling game, the input space $I$ is either Lang-attval or Lang-conj except that each element of $I$ is attached with `eos` marker. The message space $M$ is a set of discrete sequences of fixed length $k$ over a finite alphabet $A$: $M \equiv \{a_1 \cdots a_k \mid a_i \in A\}$. The goal of the game is to minimize Hamming distance between an input and an output.

**Architecture and Optimization** Speaker and listener agents are represented as a seq2seq model based on single-layer LSTMs (Hochreiter & Schmidhuber, 1997) with standard attention mechanisms (Bahdanau et al., 2015; Dong & Lapata, 2016), similarly to Chaabouni et al. (2019). As the Hamming distance is indifferentiable, we use REINFORCE (Williams, 1992) for optimization.

### 4.2 CGI FOR EMERGENT LANGUAGES

We apply CGI to emergent languages. As there is no prior knowledge on them, CGI should avoid ad hoc methods, considering the following: (1) *features in a log-linear model have to be as simple as possible*, (2) *lexical entries have to be generated automatically without any manual templates*, and (3) *lexicon size has to be minimal*; otherwise, results are hard to interpret. There is no existing method satisfying all of them simultaneously. Thus, we combine the methods of Zettlemoyer & Collins (2005), Kwiatkowski et al. (2010), and Artzi et al. (2014). For more detail, see Appendix A.

### 4.3 OTHER LANGUAGES FOR COMPARISON AND COMPOSITIONALITY METRICS

To evaluate the effectiveness of our measures, we need *less* compositional languages as well as emergent languages to apply CGI. To this end, we use *AdjSwap-x* ($x \in \{1, 2\}$). AdjSwap-$x$ is made by applying $x$-times random adjacent swaps to each message in the emergent language. As they are partially destroyed, AdjSwap-$x$ should be judged less compositional.

For compositionality metrics, we use CGF, CGL, and TopSim. When clarifying the target language, we write the metrics as (*measure*)-(*language*), e.g., TopSim-Emergent and CGF-AdjSwap-1.

---

[3]They are inspired by the commands of Chaabouni et al. (2019) and SCAN (Lake & Baroni, 2018).

$$\cfrac{\cfrac{1,1,1}{\texttt{S/S}}\quad\cfrac{16,13}{\texttt{V}}\quad\cfrac{25,1,1}{\texttt{S\textbackslash V}}}{\cfrac{\texttt{S: iter(and(rturn,run),3)}}{\texttt{S: and(iter(and(rturn,run),3),iter(and(rturn,walk),2))}}>}$$

: $\lambda x.\texttt{and}(x,\texttt{iter(and(rturn,walk),2)})$ : run : $\lambda x.\texttt{iter(and(rturn},x),3)$ <

Figure 2: Example correct derivation tree of a message "$1, 1, 1, 16, 13, 25, 1, 1$" when $(I, k, |A|) = $ (Lang-conj, 8, 31).

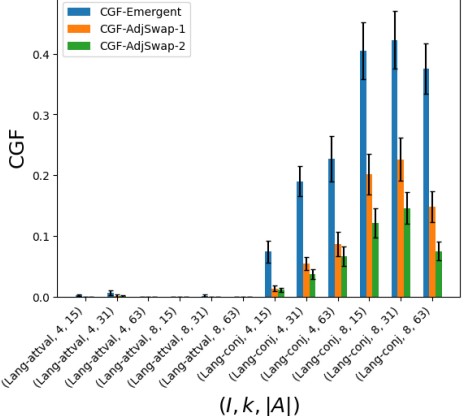

Figure 3: CGF for various $(I, k, |A|)$. The error bars represent one standard error of mean.

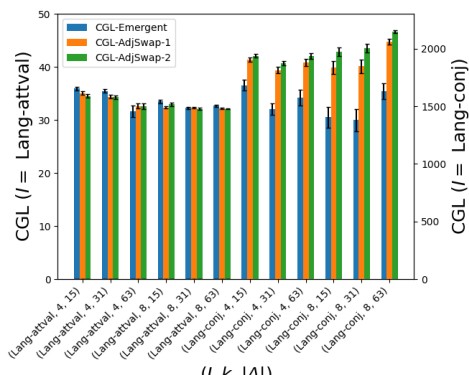

Figure 4: CGL for various $(I, k, |A|)$. The error bars represent one standard error of mean.

## 5 EXPERIMENTS AND RESULTS

In this section we show the experimental results. For (hyper)parameter settings, see Appendix B. First, Figure 2 exemplifies a derivation tree in an emergent language that CGI judges highly compositional (CGF = 0.914, CGL = 423). We can see how the message is combined to yield the meaning, which is a striking feature of CGI that the existing compositionality measures do not have.

Next, we investigate whether CGF/L works as a measure of compositionality. If CGF works, the following inequality should hold: CGF-Emergent > CGF-AdjSwap-1 > CGF-AdjSwap-2. Likewise, if CGL works, CGL-Emergent < CGL-AdjSwap-1 < CGL-AdjSwap-2. Figure 3 (resp. Figure 4) shows CGF (resp. CGL) under various $(I, k, |A|)$. For $I = $ Lang-attval, Figure 3 shows surprisingly that CGI fails: CGF-Emergent is near or equal to 0. In addition, CGL-Emergent and CGL-AdjSwap-$x$ in Figure 4 show no clear differences. Hence, neither CGF nor CGL does not recognize the compositionality of emergent languages. For $I = $ Lang-conj, Figure 3 reveals that CGF exactly shows the order of compositionality as expected: CGF-Emergent > CGF-AdjSwap-1 > CGF-AdjSwap-2. Likewise, CGL in Figure 4 shows the expected order: CGL-Emergent < CGL-AdjSwap-1 < CGL-AdjSwap-2. Hence, CGF and CGL recognize the compositionality of emergent languages.

Finally, we check the relationship between CGF/L and TopSim. We only consider the results for $I = $ Lang-conj, where CGF/L recognizes the compositionality of emergent languages. We report that TopSim and CGF show a correlation with Pearson $\rho = 0.644$ ($p = 8.77 \times 10^{-24} \ll 0.01$). Likewise, TopSim and CGL show a correlation with Pearson $\rho = -0.689$ ($p = 2.88 \times 10^{-28} \ll 0.01$). Although $\rho$s are moderate, $p$-values are considerably small. Thus, there are significant correlations between TopSim and our measures. The scatter plot between TopSim and CGF (resp. CGL) is shown in Figure 5 (resp. Figure 6) in Appendix C.

## 6 CONCLUSION AND FUTURE WORK

This paper introduces categorial grammar induction (CGI) as a new compositionality measure for the structure of emergent languages. We proposed to apply CGI to emergent languages and define two compositionality measures CGF and CGL. Our experiments revealed that CGF/L can measure

compositionality as we expected. Unlike existing measures, our approach meets compositionality in a traditional sense, allowing us to analyze emergent languages with lexical entries and derivation trees. For future work, it would be interesting to study the structure of the derivations of emergent languages. Besides, we speculate that *situated CCGs* (Artzi & Zettlemoyer, 2013) are applicable, which induce CGs considering an external world. Hence, CGI may be applicable to visual referential games as well as 2D-grid world communication.

ACKNOWLEDGEMENT

We would like to thank anonymous reviewers for helpful suggestions.

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

## A  REVIEWS AND MODIFICATIONS OF CGI

### A.1  REVIEWS ON EXISTING METHODS

Several CG induction (CGI) algorithms have been proposed. Algorithm 1 shows their common structure as a pseudo-code. Generally, the inputs to CGI are a training data $\mathcal{E} = \{(m^j, \psi^j)\}_{j=1}^N$ of message-meaning pairs, a seed lexicon $\Lambda_{\text{seed}}$, the number of iterations $T$, and a learning rate $\gamma$, while the outputs are a lexicon $\Lambda$ and a parameter $\theta$. CGI involves four procedures: (1) initialization of the lexicon and parameters (INITLEX, INITPARAM) that helps learning in early iterations, (2) update of the lexicon (UPDATELEX) that introduces a new potential lexicon, (3) update of the parameters (UPDATEPARAM) with gradient descent, and optionally (4) pruning of the lexicon (PRUNELEX) that discards a lexicon no longer in use. ZC05 (Zettlemoyer & Collins, 2005) is the first paper to formalize CGI. ZC07 (Zettlemoyer & Collins, 2007) is its improved version. In ZC05/07, INITLEX is simply $\Lambda_0 = \Lambda_{\text{seed}}$ and UPDATELEX is based on hand-crafted templates to add a new lexicon. KZGS10/11 (Kwiatkowski et al., 2010; 2011)

---

**Algorithm 1** Common Structure of CG Induction

**Input:** A dataset $\mathcal{E} = \{(m^j, \psi^j)\}_{j=1}^N$ of message-meaning pairs, a seed lexicon $\Lambda_{\text{seed}}$, the number of iterations $T$, and a learning rate $\gamma$.

**Output:** Lexicon $\Lambda$ and parameter vector $\theta$
1: $\Lambda_0 \leftarrow$ INITLEX($\mathcal{E}, \Lambda_{\text{seed}}$)
2: $\theta_0 \leftarrow$ INITPARAM($\mathcal{E}, \Lambda_{\text{seed}}$)
   $\triangleright$ Step 0: Initialize lexicon and parameter
3: **for** $t \in \{1, \ldots, T\}$ **do**
4:   $\Lambda_t^+ \leftarrow$ UPDATELEX($\mathcal{E}, \theta_{t-1}, \Lambda_{t-1}, \Lambda_0$)
     $\triangleright$ Step 1: Update Lexicon
5:   $\theta_t \leftarrow$ UPDATEPARAM($\mathcal{E}, \theta_{t-1}, \Lambda_t^+, \gamma$)
     $\triangleright$ Step 2: Update Parameter
6:   $\Lambda_t \leftarrow$ PRUNELEX($\mathcal{E}, \theta_{t-1}, \Lambda_t^+$)
     $\triangleright$ Step 3: Prune Lexicon (optional)
7: **end for**
8: **return** $\Lambda_T$ and $\theta_T$

---

modified UPDATELEX so that it can create a new lexicon by automatically merging and splitting the existing entries in use. In KZGS10/11, INITLEX returns $\mathcal{E}$ themselves with category S in addition to $\Lambda_{\text{seed}}$:

$$\Lambda_0 \leftarrow \Lambda_{\text{seed}} \cup \{m^j \vdash \texttt{S} : \psi^j \mid j = 1, \ldots, N\}$$

Then, the lexical entries are split or merged during the iteration, seeking an appropriate segmentation. A problem in KZGS10/11 is that the lexicon size increases monotonically over iterations. ADP14 (Artzi et al., 2014) addressed this issue by adding a lexicon pruning process (PRUNELEX), which discards the lexical entries that are no longer in use[4].

### A.2  MODIFICATION OF CGI

For (1), we follow ZC05 (Zettlemoyer & Collins, 2005): each feature is the count of times that each lexical entry is used in a derivation. However, ZC05 generates lexical entries with manual templates,

---

[4]ADP14 also has improvements in UPDATELEX, but we do not go into them in this paper.

contrary to (2). Instead, we follow KZGS10 (Kwiatkowski et al., 2010) which creates a new lexicon by merging and splitting existing entries in use. The problem in KZGS10 is that the lexicon size increases monotonically during iterations, which is against (3). Thus, we follow ADP14 (Artzi et al., 2014) to discard the entries no longer in use.

**INITLEX**    We set $\Lambda_{\text{seed}} = \emptyset$, as we do not have any prior knowledge on emergent languages.

**UPDATELEX**    In KZGS10, UPDATELEX includes part of a potential new lexicon pruning the rest, while ours includes all of them. This is because PRUNELEX of ADP14 would implicitly do the same thing. Moreover, the original UPDATELEX splits lexical entries as a higher-order unification problem to find $f$ and $g$ s.t. $h = f(g)$ or $h = f \circ g$, given a logical form $h$. On the other hand, ours splits the entries as a problem only to find $h = f(g)$, ensuring that $f \neq \lambda x.x.$ and $g$ is not a function.

**INITPARAM**    Since the algorithm can only search a limited space in practice, a reasonable parameter initialization is required. KZGS10 used a statistical translation method[5], while we simply compute the mean pointwise mutual information (pmi) between n-grams and the logical constants. Formally, given a feature, that is, a lexical entry $m \vdash X : \psi$, its initial parameter is defined as:

$$\frac{1}{|\text{Cnst}(\psi)|} \sum_{c \in \text{Cnst}(\psi)} \text{pmi}(m, c)$$

if $|\text{Cnst}(\psi)| > 0$ otherwise 0. $\text{Cnst}(\psi)$ enumerates the logical constants (e.g. `look`, `left`, or `1`) occurring in $\psi$.

## B    (HYPER)PARAMETERS

**Agents**    For agent architecture, the hidden state size is 100. For agent optimization, the number of mini-batches per epoch is 100, the size of mini-batches is 1000, and the learning rate is 0.001. Agents train either for 200 epochs or until loss $\mathcal{L}$ for a validation dataset reaches 0. Also, the weight of speaker's (resp. listener's) entropy regularizer $\lambda_S = 0.1$ (resp. $\lambda_L = 1$). These parameters are determined according to our preliminary experiments.

**Signaling Game**    For signaling games, an input space $I \in \{\text{Lang-attval}, \text{Lang-conj}\}$, the size $|A|$ of an alphabet $A$ is in $\{15, 31, 63\}$, and a message length $k \in \{4, 8\}$.

**CGI**    For CGI, the number of iterations $T = 10$, a learning rate $\gamma = 0.1$, and a beam size for CKY parsing is 10, referring to Artzi et al. (2014) and our preliminary experiments.

## C    CORRELATION BETWEEN TOPSIM AND CGF/CGL

We show the scatter plot between CGF-Emergent and TopSim in Figure 5. Likewise, we show the scatter plot between CGL-Emergent and TopSim in Figure 6.

---

[5]Giza++ Model 1 (Och & Ney, 2003).

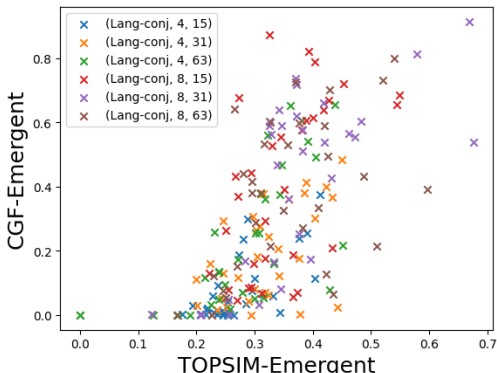
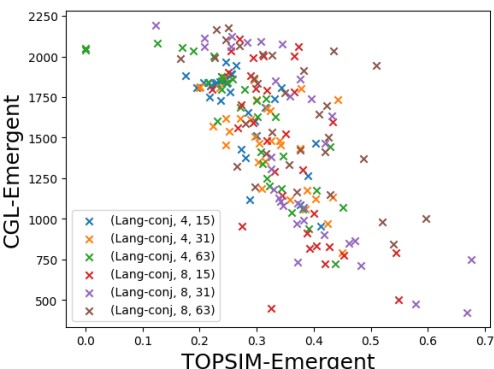

Figure 5: Scatter plot of CGF-Emergent and TopSim-Emergent.

Figure 6: Scatter plot of CGL-Emergent and TopSim-Emergent.

