# OpenReview forum: "Categorial Grammar Induction as a Compositionality Measure for Emergent Languages in Signaling Games"
_ICLR.cc/2022/Workshop/EmeCom — EmeCom Workshop at ICLR 2022_

### Official Review · Reviewer_ku5c · 2022-03-21
**Good paper**

**Rating:** Weak accept
**Confidence:** 4

**Review:**

# Summary
The paper propose to exmine the CCG grammar induced from the emergent language as a probe to measure the underlying linguistic structure, like compositionality. The authors conduct experiments on the classifical signalling game with a seq2seq LSTM model. They show that the proposed metric has some correlation with existing metric like TopoSim, and offer extra benefits.

# Strengths:
The introduction of an automatic grammar induction algorithm is a novel idea to me. Besides the grammar tree depths, I imagine there could be other interesting metrics around the induced grammar tree.

# Weakness
I would love to see more experiments on the proposed metric. To start with, is it sensitive to the grammar induction optimization process? One reason people use TopoSim is that it's simple and stable to compute. Secondly, there are some known algorithm that can improve topoSim in the classical signalling game, e.g., neural iterated learning [1], so it should be more interesting to plot this metric along side the topoSim with iterated learning.

# Final
Overally, I enjoy reading this paper, and I like the idea, while I think the experiment can be made better with the analysis suggested above.


[1] https://openreview.net/forum?id=HkePNpVKPB

---

### Official Review · Reviewer_QEmL · 2022-03-24
**Interesting approach to a complex problem**

**Rating:** Strong Accept
**Confidence:** 4

**Review:**

## summary
This paper proposes to use categorical grammars (CG) to model learned protocols in emergent communication. Inspired by work on CCGs for natural language, they use CGI to learn a lambda calculus that can model the emergent language. From there, they propose to use two metrics of the learned CG as metrics of emergent language compositionality: F1-score of the grammar on a held-out test set (CGF) and size of the CG lexicon (CGL). The idea is that if the CG better captures the learned protocol (as shown on the test set), then it will likely be a compositional protocol (CGF) and a protocol that decomposes into fewer lexical items will be more compositional (CGL).

To measure the quality of their metrics, they use LSTMs to learn to reconstruct two types of input spaces. Lang-attval which is composed of action-direction-number e.g. look-right-2, and lang-conj which is can combine two lang-attval statements with an "add" between them. The authors compare the learned languages to likely less compositional languages adjswap-{1,2} by swapping 1 or 2 tokens in the learned protocols. They find that on lang-attval, the metrics do not distinguish between the less compsitional protocols. In contrast, on lang-conj, the metrics clearly show the unswapped language to be more compositional. Furthermore, the metrics correlate with topsim providing another argument for their use.

## review

Overall, I believe the paper is interesting and very novel. To my knowledge, no one has attempted using CGs to model emergent language. Indeed, most of the current metrics for compositionality in EC do not measure non-trivial compositionality, so it is good to see more people investigate the complex ways meaning may be transmitted. Furthermore, the paper is well written and provides a great intro to CGs and overview of CGI in the appendix. I also appreciated detailing the experimental hyperparameters and showing std deviations in the graphs. I believe this paper will make for excellent discussion so I recommend it to be accepted. The following comments are mainly for the authors so that they may improve their work for future submission and perhaps give them ideas for the discussions they'd like to have.

I think the major challenge with this work is that measuring the efficacy of a compositionality metric is difficult because it requires having protocols that are less or more compositional. In TRE, Andreas shows a relationship between his metric and mutual information, human subjective opinion, topsim, and systematic generalization. In a work from last year's workshop, "Measuring non-trivial compositionality in emergent communication" Korbak et al create specific languages with common pitfalls and then demonstrate how different metrics catch different pitfalls. This paper learns a language and then uses adjswap to construct languages that are *likely* less compositional. The issue is that those languages may be worse in many other ways as well, so it isn't clear that compositionality is the exact thing your metric is measuring. Instead, I would suggest following Andreas and learning languages on a dataset and seeing which ones generalize systematically to a test set. It is likely that just changing the random seed will lead to vastly different generalization outcomes and comparing to systematic generalization would be a stronger argument than the adjswap heuristic. Alternatively, you could also specifically learn/create a protocol where TRE could not easily capture the compositionality and demonstrate your metric works better.

The other big challenge for this work is the specific metrics themselves. The idea behind F1-score and lexicon size is reasonable but they require that the learned CG is a good representation of the protocol. It becomes an issue that perhaps the reason behind a large lexicon or bad F1-score isn't the compositionality of the emergent protocol but the quality of the learned CG. You could demonstrate the CG accuracy correlates with the EC game accuracy which could help. Overall, it is a difficult thing to show because although we know natural language can be generated by something like a lambda calculus, it isn't clear that this is the sort of thing that LSTMs are outputting and so it isn't clear that CGI is accurately capturing the meaning. A qualitative analysis of the learned lexicon (something like interpretability) would be a big step towards showing this.

I would also like to point out the idea of using a CG for emergent language is quite clever and there are many other possible research ideas stemming from this. For example, you could learn a CG and use it to replace the sender's message then retrain the receiver and see if the resulting protocol is even better. Another idea is to use the CG as a loss function to guide learning a more compositional protocol. As mentioned in future work, situated CCGs would be incredibly interesting to see in a gridworld.

minor comments
- for LSTMs with attention, please also cite (Bahdanau et al, 2014)
- comparing to TRE feels like a stronger baseline than topsim although ideally you have both

---

### Decision · Program_Chairs · 2022-03-25

**Decision:**

Accept

**Comment:**

This paper takes on the difficult task of a new metric for compositionality. Both reviewers found the idea of categorical grammar novel and interesting and would like to see it pursued further. We accept this paper and look forward to discussions on this topic and future work.